# Should Patients with Waldenström Macroglobulinemia Receive a BTK Inhibitor as Frontline Therapy?

Marina Deodato *, Anna Maria Frustaci [ID], Giulia Zamprogna, Giulia Cotilli, Roberto Cairoli [ID] and Alessandra Tedeschi

Department of Hematology, Niguarda Cancer Center, ASST Grande Ospedale Metropolitano Niguarda, 20162 Milano, Italy
* Correspondence: marina.deodato@ospedaleniguarda.it; Tel.: +39-0264442668; Fax: +39-0264443019

**Abstract:** Waldenström Macroglobulinemia (WM) is a rare indolent lymphoma with heterogeneous clinical presentation. As there are no randomised trials suggesting the best treatment option in treatment-naive patients, guidelines suggest either rituximab-combining regimens or BTK-inhibitors (BTKi) as feasible alternatives. Several factors play in the decision-making process: patients' age and fitness, disease characteristics and genotype. Chemoimmunotherapy (CIT) represents a fixed-duration, less expensive and effective option, able to achieve prolonged time-to-next treatment even in patients with unfavourable genotypes. Immunosuppression and treatment-related second cancers may represent serious concerns. Proteasome-inhibitor-based regimens are effective with rapid disease control, although bortezomib-related neuropathy discourages the choice of these agents and treatment schedules may not be easily manageable in the elderly. BTKi have demonstrated high rates of response and prolonged survival together with the convenience of an oral administration and limited cytopenias. However, outcomes are impacted by genotype and some concerns remain, in particular the continuous drug exposure that may result in extra-haematological complications and drug resistance. Although next-generation BTKi have improved treatment tolerance, the question whether BTKi should be offered as frontline therapy to every patient is still debated. Giving fixed-duration schedule, prolonged time-to-next treatment and outcomes independent of genotype, CIT is still our preferred choice in WM. However, BTKi remain a valuable option in frail patients unsuitable for CIT.

**Keywords:** Waldenström Macroglobulinemia; treatment-naive; chemoimmunotherapy; proteasome inhibitors; bortezomib; BTK inhibitors; acalabrutinib; ibrutinib; zanubrutinib

## 1. Introduction

Waldenström Macroglobulinemia (WM) is a rare lymphoproliferative disorder characterized by bone marrow infiltration with monoclonal immunoglobulin M-secreting lymphoplasmacytic cells [1]. This disease usually has an indolent course, and its clinical presentation is heterogeneous, with signs and symptoms resulting from marrow or other tissue infiltration by clonal cells or from physicochemical or immunological properties of the monoclonal IgM [2].

As for other indolent lymphomas, treatment is indicated only in symptomatic patients, as guidelines recommend [3]. Common indications to treat WM include significant anaemia, thrombocytopenia, lymphadenopathy and splenomegaly; neuropathy and renal dysfunction are less frequent manifestations requiring treatment. Although rarely, some patients may require a prompt therapeutic approach to avoid irreparable organ damage or fatal complications, such as in the case of hyperviscosity syndrome [3]. Plasmapheresis is indicated in such cases to reduce IgM protein and consequently the risk of permanent organ impairment. However, the benefit of this procedure is time-limited, so systemic treatment should promptly follow [4].

Multiple therapy options are available for WM, including chemotherapy, monoclonal antibodies and proteasome inhibitors (PI). Latest insights on the disease pathophysiology have revealed that the neoplastic lymphoplasmacytic cells exhibit constitutive activation of the B-cell receptor signalling complex, of which Bruton tyrosine kinase (BTK) is a critical component [5]. Considering its crucial role in WM pathogenesis, this kinase has appeared to be a potent therapeutic target and BTK inhibitors (BTKi) have emerged as another promising option within the therapeutic landscape of WM patients [6].

Given the rarity of the disease, most of the current regimens have been adopted from data derived from phase II studies; less often, data have emerged from prospective trials enrolling patients with several types of indolent B-cell lymphomas, including lymphoplasmacytic lymphomas, while randomized trials specifically addressed to WM are even rarer [7–10]. Nevertheless, it is widely established that the selection of the optimal therapy is an individualized decision tailored by clinicians and patients together, taking into account several factors including characteristics of the patient (age, performance status, comorbidities, concomitant medications), disease (tumour burden, signs/symptoms requiring therapy), and treatment (rapidity in disease control, toxicity, management, costs) [6]. For all these reasons, no standard first-line treatment has been univocally defined.

In the last decade, also patients' genomic profiles have been demonstrated to play a significant role in providing insightful information for treatment selection. In particular, the somatic mutations in myeloid differentiation factor 88 (MYD88) and C-X-C chemokine receptor type 4 (CXCR4) have been extensively characterized [11–14].

The $MYD88^{\text{L265P}}$ mutation is present in more than 90% of WM patients and favours neoplastic cell survival and proliferation by persistently activating BCR signalling through BTK [11]. A mutation in the *CXCR4* gene can be identified in 30–40% of WM patients; this mutation is similar to the one observed in WHIM syndrome and leads to constitutive activation of the gene and its downstream signalling pathway. Two different types of *CXCR4* mutations have been identified: nonsense and frameshift mutations [12].

Based on the presence or absence of these mutations, three specific genetic groups of patients may be identified in WM: $MYD88^{\text{L265P}}/CXCR4^{\text{WT}}$ (50–60%), $MYD88^{\text{L265P}}/CXCR4^{\text{MUT}}$ (30–40%) and $MYD88^{\text{WT}}/CXCR4^{\text{WT}}$ (5–10%) [13]. These subgroups characterize different clinical presentations and, most importantly, are predictive of treatment response and survival [14].

## 2. First Line Therapies

### 2.1. Cytotoxic Agents and AntiCD20 Monoclonal Antibodies

Before the introduction of rituximab-based combinations, single-agent chemotherapy and rituximab monotherapy were historically used. One of the largest prospective randomized trials compared chlorambucil with fludarabine, demonstrating better outcomes in terms of responses and progression-free survival (PFS), as well as overall survival (OS), in patients receiving the purine analogue [7]. Furthermore, a significantly higher rate of secondary malignancies was observed in the chlorambucil arm (6-year cumulative incidence of 20.6% versus 3.7% in the fludarabine arm).

Low rates of overall response were reported with rituximab monotherapy (18–65%) translating into relatively short PFS, depending on different administration schedules [15–18]. Importantly, it should be considered that the median time to response is long and, mostly when used in monotherapy, this agent can lead to a paradoxical rise in IgM, called "IgM flare", with possible worsening of symptoms and complications secondary to hyperviscosity [15].

Compared to single agent-based treatment, regimens combining chemotherapy plus rituximab have resulted in superior response rates and sustained survival and thus have rapidly become leading players in the WM therapeutic scenario.

Purine nucleoside analogues-based regimens, including fludarabine-rituximab and fludarabine-cyclophosphamide-rituximab, showed to be highly effective in exerting prolonged PFS [19,20]. Nevertheless, their administration in the frontline has actually declined

due to a remarkable incidence of myelosuppression and immunosuppression with a high rate of infections; moreover, their use has been associated with an increased risk of secondary malignancies.

Since the publication of the study by Dimopoulos and colleagues in 2007, exploring the role of the combination of dexamethasone, rituximab and cyclophosphamide (DRC), this regimen has been widely adopted as one of the therapies of choice in treatment naïve (TN) patients, also due to its favourable toxicity profile in the elderly population [21].

DRC showed to be effective, leading to a high rate of overall response (83%) with 7% of complete remissions (CR) [22]. Median PFS was reached at 35 months and, of note, the median time to the next treatment (TTNT) was more than 4 years (51 months) [23]. The combination was well tolerated with the majority of patients (89%) completing the 6 planned cycles and with a rate of grade 3–4 infections of 12.5%. Long-term toxicity was also limited, and 10-years OS was not reached. However, it should be considered that this regimen does not allow rapid disease control as the median time to response is 4.1 months.

Bendamustine in combination with rituximab (BR) quickly emerged as an accepted standard frontline therapy after the publication of the StiL trial comparing BR versus rituximab, cyclophosphamide, doxorubicin, vincristine and prednisone (R-CHOP) in patients with untreated indolent lymphomas, including 41 with lymphoplasmacytic lymphomas/WM [8]. BR demonstrated to be superior to R-CHOP in terms of PFS (median PFS 69.5 months with BR versus 28.1 months with R-CHOP, $p$ = 0.0033). Although bendamustine was administered at the dosage of 90 mg/m$^2$ on days 1–2, the regimen showed to be better tolerated compared to R-CHOP with a rate of all grade infections of 37% versus 50%. The efficacy of the BR regimen has been confirmed in other retrospective series. In the trial of the French Innovative Leukaemia Organization, it emerged that the reduction of bendamustine dosage or the number of cycles (44% of patients) did not influence the outcome in terms of PFS [24].

The impact of genotype on BR treatment in TN patients has been evaluated by Zanwar et al. [25]. The very good partial response (VGPR) rates were comparable between patients with *MYD88*$^{L265P}$ and *MYD88*$^{WT}$ genotypes (41% and 50%, respectively, $p$ = 0.55), and the 4-year PFS was 71% for both groups. Differently, patients harbouring *CXCR4* mutation showed a numerically lower rate of ≥VGPR (33% versus 57% for those with *CXCR4*$^{WT}$ genotype, $p$ = 0.3) and a trend towards shorter PFS.

No prospective studies have directly compared BR with DRC; however, these two regimens were retrospectively analysed in a population of 160 patients (67 TN) treated at Mayo Clinic [26]. In the TN setting, the overall response rate (ORR) was similar between the two regimens, but the median time to best response was shorter in the BR cohort (6.1 versus 11 months with DRC). Moreover, although not statistically significant, the 2-year PFS was superior in the BR group (88 versus 61% in the DRC arm, $p$ = 0.07), without an increase in toxicity. The *MYD88* mutation status was available in only 48 patients of the entire cohort and did not result to have an impact on outcomes of both BR and DRC. *CXCR4* mutations were not evaluated.

Despite some series reported positive outcomes for autologous stem cell transplant (ASCT) for WM in first-line setting [27–30], the lack of comparative trials made it difficult to provide strong evidence and high-quality recommendations on this topic. This became truer after the availability of BTKi.

Nevertheless, guidelines do not recommend frontline ASCT outside clinical trials, unless there are other indications such as amyloidosis or transformation to an aggressive lymphoma [6,31,32].

Considering the possibility of multiple relapses, the use of frontline stem cell toxic regimes in younger patients potentially candidates to salvage ASCT has been usually avoided [6].

### 2.2. Proteasome Inhibitor-Based Therapy

Combination strategies with bortezomib have shown significant activity in the treatment of WM.

Several trials have addressed the role of bortezomib plus rituximab with or without dexamethasone. The combinations allowed for achieving a high rate of responses (85–96%) with a very short time to median response (1.4 months–3.7 months) [33–37]. Progression-free survival showed to be longer in the study of Treon et al., in which four cycles of induction were followed by four cycles of maintenance (median 66 months) compared to the only five courses considered in the European study (median 43 months). Peripheral neuropathy resulted a common AE, leading to a high rate of therapy discontinuation (60%) when the PI was administered twice weekly. Importantly, a significant reduction in neurological toxicity was obtained with a weekly administration of bortezomib. Concern remains about whether dexamethasone should be added to the combination, considering that the indirect comparison showed similar outcomes with or without the use of this agent.

The question of whether bortezomib-based regimens may be superior to the most common used CIT regimens was explored in three comparative trials of previously untreated patients.

BR, DRC and BDR were retrospectively compared by Abeykoon et al. in 220 cases [38]. BR resulted superior to both DRC and BDR in terms of ORR (98% versus 78% versus 84%; respectively, $p = 0.003$), MRR (95% versus 53% versus 68%, respectively, $p < 0.0001$) and median time to best response (4.5 months versus 5.9 months versus 6.7 months, respectively, $p = 0.005$). Treatment with BR was also associated with a better PFS (median 5.2 years vs. 4.3 years with DRC and 1.8 years with BDR; $p = 0.0003$), even though OS was similar across the three groups.

Differently, no differences in terms of response rates were observed by Castillo et al. when the three regimens were retrospectively compared [39]; however, a trend towards a better PFS was recorded in BR-treated patients with again no differences in terms of OS.

The addition of bortezomib to DRC (B-DRC) did not translate into a PFS advantage compared to DRC in the Multicenter European Phase II randomized trial [9]. Importantly, the study confirmed that genotype did not influence outcomes in both treatment arms. Despite a higher occurrence of peripheral neuropathy in patients receiving B-DRC, rates of grade $\geq 3$ AEs were comparable.

Second-generation neuropathy-sparing PI carfilzomib, administered in association with rituximab and dexamethasone for six induction plus six maintenance cycles, allowed to achieve a good quality of responses independent of *MYD88* and *CXCR4* mutational status [40]. Median PFS was achieved at 46 months. Importantly, only one grade 2 peripheral neuropathy (3.2%) occurred, with no grade 3–4 events.

Similarly, oral PI ixazomib in combination with dexamethasone and rituximab showed low rates of neuropathy. ORR was high and median PFS was not reached due to the short follow-up of the study [41].

Table 1 summarises the main regimens employed in previously untreated WM patients.

**Table 1.** Summary of main regimens employed in previously untreated WM patients.

| Author | Regimen | N. pts | Responses (%) | Survival Rates | Genotype Impact | F/U (m) |
|---|---|---|---|---|---|---|
| Leblond 2013 [7] | Chlorambucil 8 mg/m$^2$ daily for 10 d every 28 d vs. Fludarabine 40 mg/m$^2$ daily for 5 d every 28 d | 170 169 | ORR 36% ORR 46% | mPFS 27 m mPFS 38 m | NA | 36 m |
| Treon 2009 [19] | R + Fludarabine: R 375 mg/m$^2$/w at w 1 to 4, 17, 18, 30, 31 + 6 cycles of F 25 mg/m$^2$ daily for 5 d at w 5, 9, 13, 19, 23, and 27 | 43 (27 TN) | ORR 95% MRR 86% CR + VGPR% 37% | mPFS 77.6 m | NA | 40.3 m |

**Table 1.** *Cont.*

| Author | Regimen | N. pts | Responses (%) | Survival Rates | Genotype Impact | F/U (m) |
|---|---|---|---|---|---|---|
| Tedeschi 2012 [20] | FCR: R 375 mg/m$^2$ on d 1 + F 25 mg/m$^2$ and C 250 mg/m$^2$ on d 2–4, every 28 d for 6 cycles | 43 (28 TN) | ORR 79% MRR 74% CR + VGPR% 32% | mPFS NR 2 y OS 88.4% 4 y OS 69.1% | NA | 37.2 m |
| Kastritis 2015 [23] | DRC: D 20 mg + R 375 mg/m$^2$ on d 1 + C 100 mg/m$^2$ bid on d 1–5, every 21 d for 6 cycles | 72 | ORR 83% MRR 74% CR 7% | mPFS 35 m mOS 95 m | NA | 7 y |
| Rummel 2013 [8] | BR: Bendamustine 90 mg/m$^2$ on d 1, 2 + R 375 mg/m$^2$ on d 1, every 4 w for 6 cycles | 22 | ORR 93% CR 40% | mPFS 69.5 m | NA | 45 m |
| Laribi 2018 [24] | BR: Bendamustine 90 mg/m$^2$ on d 1, 2 + R 375 mg/m$^2$ on d 1, every 4 w for 6 cycles | 69 | ORR 97% MRR 96% CR + VGPR 56% | 2 y PFS 87% 2 y OS 97.1% | *MYD88*$^{L265P}$ vs. *MYD88*$^{WT}$: ORR, PFS: NS *CXCR4*$^{MUT}$ vs. *CXCR4*$^{WT}$: ORR; PFS: NS | 23 m |
| Zanwar 2022 [25] | BR: Bendamustine 90 mg/m$^2$ on d 1, 2 + R 375 mg/m$^2$ on d 1, every 4 w for 6 cycles | 208 | ORR 95% MRR 93% VGPR 31% | Est mPFS 5.9 y Est 5 y OS 90% | *MYD88*$^{L265P}$ vs. *MYD88*$^{WT}$: ORR, PFS: NS *CXCR4*$^{MUT}$ vs. *CXCR4*$^{WT}$: ORR; PFS: NS | 4 y |
| Paludo 2018 [26] | BR: Bendamustine 90 mg/m$^2$ on d 1, 2 + R 375 mg/m$^2$ on d 1, every 4 w for 6 cycles vs. DRC: D 20 mg + R 375 mg/m$^2$ on d 1 + C 100 mg/m$^2$ bid on d 1–5, every 21 d for 6 cycles | 60 (17 TN) 100 (50 TN) | ORR 93% MRR 86% VGPR 29% ORR 96% MRR 87% VGPR 17% | 2 y PFS 88% 2 y PFS 61% | *MYD88*$^{L265P}$ vs. *MYD88*$^{WT}$: ORR, PFS: NS | 30 m |
| Treon 2015 [34] | BDR: Bortezomib 1.3 mg/m$^2$ IV + D 40 mg on d 1, 4, 8, 11 + R 375 mg/m$^2$ on d 11. A total of 4 cycles of induction + 4 cycles, each 3 months apart, for maintenance | 23 | ORR 96% MRR 91% CR + VGPR 52% | 5 y PFS 57% 5 y OS 95% | NA | 8.5 y |
| Ghobrial 2010 [35] | Bortezomib + R: Bortezomib 1.6 mg/m$^2$ IV on d 1, 8, 15, every 28 d for 6 cycles + R 375 mg/m$^2$/w on cycles 1 and 4 | 26 | ORR 89% MRR 66% CR + VGPR 8% | 1 y EFS 79% | NA | 14 m |

| Author | Regimen | N. pts | Responses (%) | Survival Rates | Genotype Impact | F/U (m) |
|---|---|---|---|---|---|---|
| Gavriatopoulou 2017 [37] | BDR: Bortezomib 1.3 mg/m$^2$ IV on d 1, 4, 8, 11 (cycle 1), B 1.6 mg/m$^2$ IV on d 1, 8, 15, 22 (cycles 2–5) + D 40 mg and R 375 mg/m$^2$/w in cycle 2 and 5. Every 35 d for 5 cycles | 59 | ORR 85% MRR 68% CR + VGPR 10% | mPFS 43 m 7 y OS 66% | NA | 86 m |
| Abeykoon 2021 [38] | BR: Bendamustine 90 mg/m$^2$ on d 1, 2 + R 375 mg/m$^2$ on d 1, every 4 w for 6 cycles vs. BDR: Bortezomib 1.3 mg/m$^2$ IV on d 1, 4, 8, 11 (cycle 1), B 1.6 mg/m$^2$ IV on d 1, 8, 15, 22 (cycles 2–5) + D 40 mg and R 375 mg/m$^2$/w in cycle 2 vs. DRC: D 20 mg + R 375 mg/m$^2$ on d 1 + C 100 mg/m$^2$ bid on d 1–5, every 21 d for 6 cycles | 83 45 92 | ORR 98% MRR 96% ORR 84% MRR 68% ORR 78% MRR 53% | mPFS 5.2 y 4 y OS 90% mPFS 1.8 y 4 y OS 87% mPFS 4.3 y 4 y OS 87% | *MYD88*$^{L265P}$ vs. *MYD88*$^{WT}$: ORR, PFS: NS | 2.3 y 5.2 y 6.3 y |
| Castillo 2018 [39] | BR: Bendamustine 90 mg/m$^2$ on d 1, 2 + R 375 mg/m$^2$ on d 1, every 4 w for 6 cycles vs. BDR: bortezomib 1.6 mg/m$^2$ + D 20 mg on d 1, 4, 8 and 11 or 1, 8, 15 and 22, every 3 and 4 w, respectively, + rituximab 375 mg/m$^2$ on d 11 or 22, respectively, for 4 cycles vs. DRC: C 1000 mg/m$^2$ + plus D 20 mg + R 375 mg/m$^2$ on d 1, for 6 cycles | 57 87 38 | ORR 98% MRR 94% CR + VGPR 45% ORR 90% MRR 83% CR + VGPR 35% ORR 89% MRR 84% CR + VGPR 42% | mPFS 5.5 m 10 y OS 95% mPFS 5.8 m 10 y OS 96% mPFS 4.8 m 10 y OS 81% | NA | 3 y 4 y 5 y |
| Buske 2020 [9] | B-DRC: DRC + Bortezomib sc 1,6 mg/m$^2$ on d 1, 8, 15, every 28 d for 6 cycles vs. DRC: D 20 mg + R 375 mg/m$^2$ on d 1 + C 100 mg/m$^2$ bid on d 1–5, every 21 d for 6 cycles | 100 100 | ORR 91% MRR 79% CR + VGPR 19% ORR 87% MRR 69% CR + VGPR 11% | mPFS NR mPFS 50.1 m | *MYD88*$^{L265P}$ vs. *MYD88*$^{WT}$: ORR, PFS: NS *CXCR4*$^{MUT}$ vs. *CXCR4*$^{WT}$: ORR; PFS: NS | 27.5 m |

**Table 1.** *Cont.*

| Author | Regimen | N. pts | Responses (%) | Survival Rates | Genotype Impact | F/U (m) |
|---|---|---|---|---|---|---|
| Meid 2017 [40] | CaRD: carfilzomib 20 mg/m$^2$ (cycle 1) and 36 mg/m$^2$ (cycles 2 and beyond), D 20 mg on d 1, 2, 8, 9; Rituximab 375 mg/m$^2$ on d 2, 9 of each 21-d cycle. Six induction cycles +8 cycles, each 8 w apart for maintenance | 31 | ORR 81% MRR 71% CR + VGPR 39% | mPFS 46 m | *MYD88*$^{L265P}$ vs. *MYD88*$^{WT}$: ORR: NS *CXCR4*$^{MUT}$ vs. *CXCR4*$^{WT}$: ORR: NS | NA |
| Castillo 2018 [41] | IDR: ixazomib 4 mg and D 20 mg on d 1, 8, and 15 every 4 w for cycles 1 and 2, + R 375 mg/m$^2$ on d 1, e 4 weeks for cycles 3 to 6. Then 6 cycles, each 8 w apart, for maintenance | 26 | ORR 96% MRR 77% | mPFS NR | *CXCR4*$^{MUT}$ vs. *CXCR4*$^{WT}$: ORR: NS | 22 m |

R: rituximab; D: dexamethasone; C: cyclophosphamide; F: fludarabine; ORR: overall response rate; MRR: major response rate; VGPR: very good partial remission; CR: complete remission; PFS: progression-free survival; OS: overall survival; EFS: event-free survival; mPFS: median progression-free survival; mOS: median overall survival; NR: not reached; NA: not available; NS: not significant; TN: treatment-naïve; d: days; w: weeks; m: months; y: years; n: number; pts: patients; F/U: follow-up; vs: versus.

### 2.3. Rituximab Maintenance

Limited and not univocal evidence has addressed the role of maintenance with rituximab in WM. In the previously mentioned retrospective study by Castillo and colleagues, patients receiving prolonged rituximab administration as maintenance achieved higher ORR (97% versus 68%), longer median PFS (6.8 years versus 2.8 years) and better 10-year OS rate (84% versus 66%) compared to those discontinuing treatment after the induction phase [39]. In the MAINTAIN trial patients received induction therapy with BR and were then randomised to observation or rituximab every 2 months for 2 years [42]. The study confirmed the high efficacy of induction therapy with BR and did not demonstrate an advantage of rituximab maintenance in terms of OS or PFS. Moreover, both studies highlighted the higher rate of infections in the cohort receiving the antiCD20 monoclonal antibody with prolonged schedule.

### 2.4. BTK Inhibitors

Ibrutinib, at a dosage of 420 mg daily until disease progression or unacceptable toxicity, was the first drug that FDA and EMA specifically approved for the treatment of WM [43,44].

After the first report in the relapse/refractory (R/R) setting [45], ibrutinib was explored by Treon and colleagues in 30 untreated WM patients [46]. Considering the poor outcome of *MYD88*$^{WT}$ patients treated with ibrutinib, only *MYD88*$^{MUT}$ patients were enrolled in this study. With the longer follow-up of 50.1 months, overall and major response rates were 100% and 87%, respectively; none of the patients achieved a CR [47]. The median time for minor and major responses was 1 and 1.9 months, respectively. The median PFS was not reached and the 4-year PFS rate was 76%. No deaths were recorded during active ibrutinib treatment, for an OS rate of 100%. Most of the AEs were mild, while grade $\geq$ 2 atrial fibrillation occurred in 20% of patients. About 10% of patients had to reduce ibrutinib dosage or discontinue treatment, respectively, due to adverse events. Importantly, one ventricular fibrillation was reported. When patients were stratified according to *CXCR4* mutational status, *CXCR4*$^{MUT}$ patients (47% of the whole population) showed numerical lower, despite not significant, rate of VGPR, a significantly longer median time to major response and a sixfold increased risk of progression or death (HR 6.03; 95% CI: 0.7–51.6; $p = 0.09$).

In the phase III iNNOVATE trial, the addition of ibrutinib to rituximab resulted in higher ORR, MRR, and PFS compared to rituximab monotherapy even in the TN patients enrolled in the study [48]. Of note, the addition of rituximab to ibrutinib was able to abrogate the negative impact of *CXCR4* mutational status. Moreover, while response rates were slightly inferior in cases not harbouring the $MYD88^{\text{L265P}}$ mutation, these minor differences did not affect the PFS benefit.

The second-generation BTKi acalabrutinib was developed to be more selective than ibrutinib. Only 14 TN patients received acalabrutinib in a phase II multicenter study [49]. As with ibrutinib, the majority of patients achieved a response (93%) with no VGPR or CR. Estimated 66-months PFS and OS were 84% and 91%, respectively. In this study, patients were not stratified according to *CXCR4* mutational status. Acalabrutinib was discontinued in 50% of patients, AEs being the main reason. Grade 1–2 atrial fibrillation and haemorrhage were reported in 7 and 71%, with no grade 3–4 events.

Zanubrutinib, a potent irreversible next-generation BTKi, was first evaluated in a phase I/II trials enrolling subjects with B-Cell lymphoid malignancies [50]. In the cohort of 24 TN WM patients, overall and major response rates resulted in 100 and 87.5%, respectively. The median time to major response was 2.8 months, and the quality of response increased while on treatment. The 2-year estimated EFS was 91.5%. Among AEs of special interest, 62.3% of subjects experienced bleeding events, mostly of grade 1–2; atrial fibrillation was reported only in 5.2% of patients.

Based on these promising results, the phase III ASPEN trial was designed as a head-to-head comparison of zanubrutinib to ibrutinib [10]. Of 201 patients enrolled, 37 were TN WM unsuitable for standard CIT. Considering the inferior activity of ibrutinib on $MYD88^{\text{WT}}$ cases, only patients with *MYD88* mutation were randomised, while in a second cohort $MYD88^{\text{WT}}$ patients were directly assigned to zanubrutinib.

At the first follow-up of the study, categorical responses of TN patients were similar between the two BTKi. The follow up was too short to draw conclusions on survival outcomes. In the 43 months follow-up of the study, only aggregate results were considered, and outcomes were not categorised according to treatment status [51]. Patients treated with zanubrutinib achieved deeper responses with a shorter time to reach VGPR (5.6 and 22.1 months, respectively, *p* = 0.35). Progression-free survival and OS were still not reached. Importantly, longer follow-up highlighted the better efficacy of zanubrutinib toward *CXCR4* mutated patients. The incidence of BTKi-related AEs was lower with zanubrutinib. Only neutropenia occurred more frequently in the next-generation BTKi arm, not leading to a higher incidence of grade 3–4 infections. Of note, patients receiving zanubrutinib showed an inferior rate of discontinuation and dose reductions.

In cohort 2, including 28 $MYD88^{\text{WT}}$ patients (5 TN), treatment with zanubrutinib led to a remarkable 80% ORR, with 40% MRR and 20% of VGPR [52]. After a median follow-up of 43 months, PFS and OS rates were 53.8% and 83.9%, respectively [51].

No prospective trials have directly compared BTKi versus CIT. Only a retrospective study on 246 $MYD88^{\text{MUT}}$ patients, comparing BR and ibrutinib, at 4.2 years of follow-up showed similar PFS and OS rates, with deeper responses attained in the CIT cohort [53].

Clinical trial data of BTKi in untreated WM patients are summarised in Table 2.

**Table 2.** Summary of clinical trials data of BTKi in untreated WM patients.

| Author | BTKi | N. pts | Responses | Survival Rates | Genotype Impact | F/U |
|---|---|---|---|---|---|---|
| Castillo 2022 [47] | Ibrutinib 420 mg daily | 30 | ORR 100% MRR 87% VGPR 30% | 4 y PFS 76% | longer TTMR in $CXCR4^{\text{MUT}}$ vs. $CXCR4^{\text{WT}}$ | 50.1 m |

**Table 2.** *Cont.*

| Author | BTKi | N. pts | Responses | Survival Rates | Genotype Impact | F/U |
|---|---|---|---|---|---|---|
| Buske 2022 [48] | ibrutinib 420 mg daily + R 375 mg/m$^2$ on d1 of w 1–4, 17–20 vs. placebo + R 375 mg/m$^2$ on d1 of w 1–4, 17–20 | 75 (45 TN) 75 (45 TN) | ORR 91% MRR 76% CR + VGPR: 27% ORR 53% MRR 41% CR + VGPR 9% | 48 m PFS 70% 48 m PFS 32% | $MYD88^{L265P}$ vs. $MYD88^{WT}$: ORR, PFS: NS $CXCR4^{MUT}$ vs. $CXCR4^{WT}$: ORR; PFS: NS | 50 m |
| Owen 2022 [49] | Acalabrutinib 100 mg bid | 106 (14 TN) | ORR 93% MRR 79% | 66 m PFS 84% 66 m OS 91% | NA | 63.7 m |
| Trotman 2020 [50] | zanubrutinib 160 mg bid | 77 (24 TN) | ORR 100% MRR 87.5% CR + VGPR 33% | Est 2 y EFS 92% | $MYD88^{L265P}$ vs. $MYD88^{WT}$: ORR, PFS: NS $CXCR4^{MUT}$ vs. $CXCR4^{WT}$: ORR; PFS: NS | 23.5 m |
| Tam 2020 [10] | ibrutinib 420 mg daily vs. zanubrutinib 160 mg bid | 99 (18 TN) 102 (19 TN) | ORR 89% MRR 67% CR + VGPR 14% ORR 95% MRR 74% CR + VGPR 26% | mPFS NR Est 18 m OS 97% mPFS NR Est 18 m OS 93% | $CXCR4^{MUT}$ vs. $CXCR4^{WT}$: ORR: NS | 18.5 m |
| Dimopoulos 2022 [51] | ibrutinib 420 mg daily vs. zanubrutinib 160 mg bid | 99 (18 TN) 102 (19 TN) | All pts CR + VGPR 22% All pts CR + VGPR 36% | All pts mPFS NR mOS NR All pts mPFS NR mOS NR | $CXCR4^{WT}$ Ibr CR + VGPR 28% Zanu CR + VGPR 45% (*p* = 0.04) $CXCR4^{MUT}$ ibr CR + VGPR 5% Zanu CR + VGPR 21% (*p* = 0.15) * | 43 m |
| Dimopoulos 2020, 2022 [51,52] | zanubrutinib 160 mg bid | 28 (5 TN) | ORR 80% MRR 40% VGPR 20% | All pts 43 m PFS 53.8% 43 m OS 83.9% | only $MYD88^{WT}$ pts | 43 m |
| Abeykoon 2022 [53] | BR: Bendamustine 90 mg/m$^2$ on d 1, 2 + R 375 mg/m$^2$ on d 1, every 4 w for 6 cycles Vs. ibrutinib 420 mg daily | 208 139 | ORR 94% MRR 92% CR + VGPR 50% ORR 94% MRR 83% CR + VGPR 33% | 4 y PFS 73% 4 y OS 94% 4 y PFS 73% 4 y OS 82% | NA | 4.2 y |

BTKi: BTK inhibitor; R: rituximab; N: number; pts: patients; F/U: follow-up; TTMR: time to major response; est: estimated; ORR: overall response rate; MRR: major response rate; VGPR: very good partial remission; CR: complete remission; PFS: progression-free survival; OS: overall survival; EFS: event-free survival; mPFS: median progression-free survival; mOS: median overall survival; NR: not reached; NA: not available; NS: not significant; TN: treatment naïve; d: days; w: weeks; m: months; y: years; vs: versus. * NB: outcomes not categorised according to treatment status.

### 3. Making a Choice in First Line

National Comprehensive Cancer Network (NCCN), and European Society for Medical Oncology (ESMO) guidelines, suggest both BTKi and rituximab-based combinations as treatment options for TN patients [32,54]. Importantly, there are some differences in BTKi approval between the US and Europe. In fact, according to EMA authorisation, only untreated WM patients unsuitable for CIT should receive BTKi [44].

In recent years in another indolent lymphoproliferative disorder, chronic lymphocytic leukaemia, BTKi have commonly replaced CIT as frontline treatment, based on phase III trials demonstrating the superiority and better tolerability of these agents regardless of patients' fitness and genomic profile [55–57]. Differently, no prospective randomised trials comparing BTKi to CIT have been performed, to better address treatment decisions in WM. That said, clinicians have to rely on different tools in order to identify the best therapeutic strategy. Patient-related factors, such as fitness status, comorbidities and concomitant medications, defined as prescribed drugs assumed on a regular basis [58], have a central role in the decision-making process. Disease factors, such as clinical manifestations and tumour burden, should drive initial therapy selection based also on the urgency needed to achieve disease control. Furthermore, in recent years, cumulative evidence in the literature supports the importance of consider genotype when selecting the best treatment options.

Rituximab-based combinations, in suitable patients, are considered as preferred regimens according to several guidelines [6,32] and the most commonly used systemic therapies in clinical practice [59]. Purine analogues use has been discouraged, not only due to the high rate of infections, but also the significant incidence of secondary malignancies.

Bortezomib-based regimens have shown remarkable responses and prolonged survival in prospective trials, with the advantage of a chemo-free mechanism of action. However, their role as frontline therapy is still limited as comparative studies, including a randomised trial, failed to demonstrate the superiority of bortezomib-combinations over CIT. Peripheral neuropathy, moreover, represents additional and often irreversible toxicity, that may discourage the choice of these regimens. Although the new proteasome inhibitors is are better tolerated, the prolonged maintenance makes these schedules not easily manageable in WM, especially considering the advanced age of this population.

Bendamustine plus rituximab is currently one of the most commonly prescribed treatments in clinical practice, and the one we use most frequently in our institution. The regimen enables to obtain rapid and profound disease control, translating into prolonged PFS and TTNT. To avoid myelosuppression and immunosuppression, mainly in elderly patients, bendamustine dosage may be reduced or less than six cycles may be administered; this is not detrimental to outcomes [24]. Although this regimen requires intravenous administration, the fixed-duration schedule plays in favour of this combination. Another factor in favour of BR choice is that its efficacy is independent of poor prognostic genotypes such as $MYD88^{WT}$ and $CXCR4^{MUT}$.

An alternative option for more unfit patients may be the administration of DRC, due to the limited rate of adverse events. With this easy-to-administered regimen with a fixed-duration schedule, a long time to next treatment (>4 years) may be achieved. Nevertheless, delayed time to response, more than 4 months, limits its use to low-tumour burden diseases. In our opinion, DRC may have a key role in patients with low-tumour burden in which the predominant symptoms are secondary to IgM-related immunologic disorders.

The main concern with the use of cytotoxic agents in the first line is the development of secondary malignancies. Data on the incidence of BR or DRC-related neoplasia in the literature are not conclusive, also considering that patients with WM are at higher risk of secondary malignancies independently of treatment status [60].

With longer follow-ups from prospective trials, the activity of BTKi in WM is becoming more and more evident. Nevertheless, it is important to highlight that the great majority of the literature data refers to the R/R setting, with the number of previously untreated patients being limited [10,46,49]. Almost all patients achieve a response with these agents, with the quality of responses ameliorating over time. Furthermore, in spite of a relatively

low rate of CRs, progressions are rare even at more than 4 years of observation [47,49]. Despite such favourable evidence, various factors should be considered before selecting a BTKi as frontline therapy.

First of all, genotype has a clear influence on treatment outcomes. $MYD88^{WT}$ patients should not receive a BTKi as the first line. Historical data showed the negative impact of $MYD88^{WT}$ status on ibrutinib so these patients were excluded from the phase II TN trial [46]. Zanubrutinib has shown to be more effective than ibrutinib in this setting of poor prognostic genotype, although PFS is shorter than the one observed in patients carrying $MYD88^{L265P}$ mutation. In regards to *CXCR4* mutational status, $CXCR4^{MUT}$ patients treated with ibrutinib show a significantly longer time to response, worse quality of response and a trend towards inferior PFS compared to $CXCR4^{WT}$ patients, thus limiting the use of this inhibitor in this setting. Despite $CXCR4^{MUT}$ patients achieving superior outcomes with zanubrutinib, this population still presents a longer time to VGPR and a lower rate of deep responses compared to $CXCR4^{WT}$ cases. Larger studies with longer follow-ups, analysing the role of zanubrutinib in the first line, are needed to better define its role in *CXCR4* mutated patients.

BTK inhibitors have the convenience of an oral administration and a low rate of cytopenias compared to both CIT and PI-based regimens. Nevertheless, we are dealing with an indefinite-time therapy that implies continuous drug exposure, with a consequently higher risk of extra-hematologic complications leading to dose reductions or discontinuation. In the only available retrospective trial comparing the outcomes of continuous ibrutinib versus BR, discontinuations due to AEs were higher with the BTKi (33% versus 13%) [53]. The BTKi-related AEs of special interest, in particular, atrial fibrillation and haemorrhages, may be of concern in the elderly population with comorbidities and multiple concomitant medications. Patients assuming antiplatelets and anticoagulants, exposed to an increased risk of bleeding events, should be carefully evaluated, as well as those receiving drugs inhibiting the CYP3A4 system, that may increase BTKi plasma levels with a potential excess of toxicity [61]. Furthermore, the BTKi dosage should be adjusted according to the potency of induction or inhibition exerted by CYP3A4 interacting drugs [61]. It should be reminded that BTKi-related adverse events of special interest are mitigated by the higher selectivity of next-generation BTKi.

An indefinite therapy is also associated with the risk of drug resistance development and consequently the need for salvage treatment. Limited data are available on the role of CIT after BTKi failure [62]. Moreover, new target agents such as venetoclax or non-covalent BTKi, known to be effective even in the presence of $BTK^{CYS481}$ and PLC$\gamma$2 mutations, are currently commercially unavailable.

Compliance with continuous treatment is also an aspect that we should consider when selecting BTKi therapy. Not lastly, the financial toxicity of continuous treatment remains an important issue.

In our opinion, since the majority of patients in clinical practice are suitable for rituximab-containing regimens, we consider BR and DRC as the treatments of choice in the first line. Chemoimmunotherapy has in fact the advantages of a fixed-duration schedule, high response rates independent of genotype and prolonged time-to-next treatment. Safety profile, even when balancing long-term toxicities, still remains acceptable.

Adopting this strategy, the use of BTKi could be reserved for the salvage setting.

Nevertheless, in a small proportion of cases, DRC and more so BR could be not tolerated. Although there is not a univocal definition of fitness in WM, advanced age and comorbidity burden may lead physicians to select a BTKi as primary therapy. In this population, considering that zanubrutinib is characterised by higher selectivity, translating into better tolerability, and efficacy even in high-risk genotypes, this agent should be preferred.

Figure 1 summarises the pros and cons of treatment options in Waldenström Macroglobulinemia.

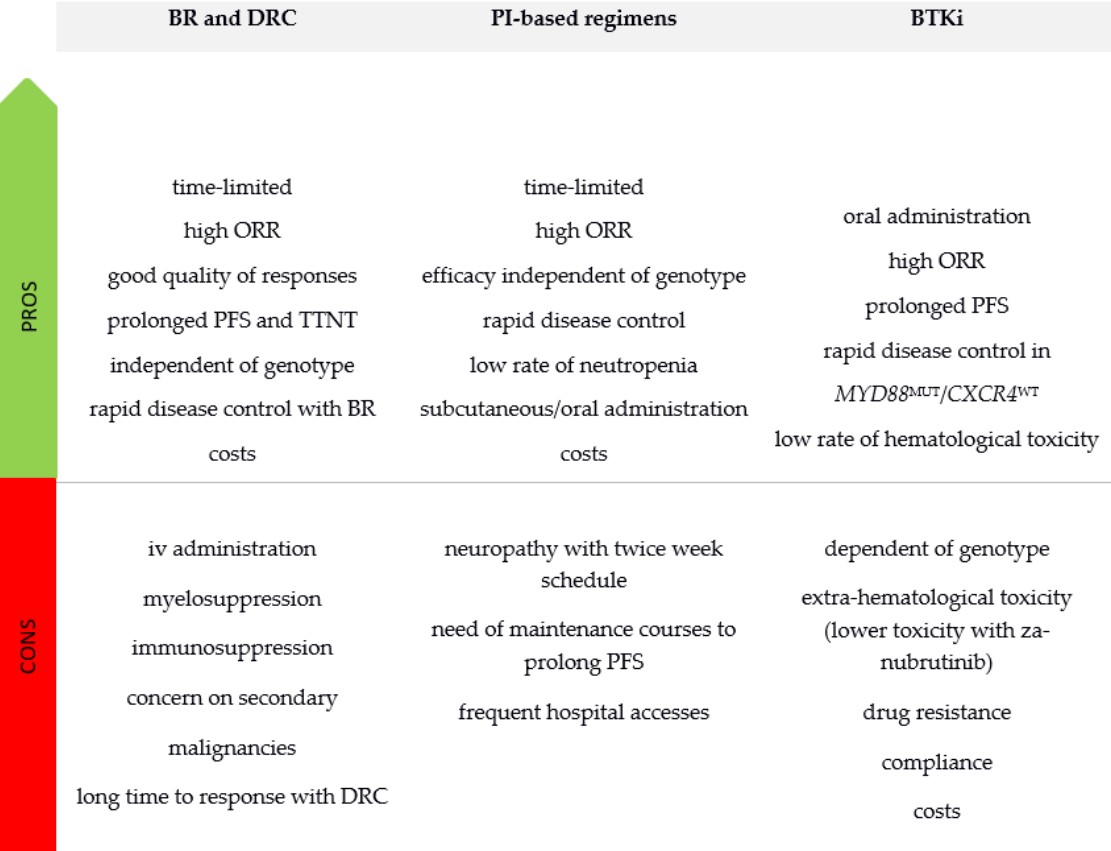

**Figure 1.** Pros and cons of treatment options in Waldenström Macroglobulinemia. BR: bendamustine+rituximab; DRC: dexamethasone+rituximab+cyclophosphamide; PI: proteasome inhibitor; BTKi: BTK inhibitors; ORR: overall response rate; PFS: progression-free survival; TTNT: time to next treatment; iv: intravenous.

**Author Contributions:** M.D.: writing—original draft; A.M.F.: writing—review and editing, supervision; G.Z.: writing—review and editing; G.C.: writing—review and editing; A.T.: conceptualisation, writing—review and editing, validation, supervision; R.C.: supervision. All authors have read and agreed to the published version of the manuscript.

**Funding:** This research received no external funding.

**Institutional Review Board Statement:** Not applicable.

**Informed Consent Statement:** Not applicable.

**Data Availability Statement:** Data sharing is not applicable to this article.

**Conflicts of Interest:** The authors declare no conflict of interest.

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
