# Peer review of "Should Patients with Waldenström Macroglobulinemia Receive a BTK Inhibitor as Frontline Therapy?"

_hemato, doi:10.3390/hemato3040046_

Round 1
Reviewer 1 Report
Wery good review of the literature. I have only one question:
Is there any role of auto. HSCT/high dose CT in WM? (Comparing endless BTKi)
Author Response
Thank you for your comment on our manuscript.
Below is our response to each point raised.
1) Is there any role of auto. HSCT/high dose CT in WM? (Comparing endless BTKi)
In our paper we did not focus on ASCT as the aim was to discuss the role of BTKi in first line. For this reason, we considered to avoid a chapter specifically addressing ASCT. We added a comment on ASCT in the immunochemotherapy chapter.
Reviewer 2 Report
First line therapies in WM are well summarized and discussion on decision-making process appropriate.
Minor remarks :
- sentences on chemoimmunotherapy as the preferred choice for frontline therapy in WM in the abstract and in paragraph 2 should be formulated to reflect the opinion of the authors
- table 2, Dimopoulos study on MYD88WT should be reference n°45 and the date 2020 ?
- figure 1: should be "CONS" and not "CON" on the left
Author Response
Thank you for your comments on our manuscript.
Below is our response to each point.
1) Sentences on chemoimmunotherapy as the preferred choice for frontline therapy in WM in the abstract and in paragraph 2 should be formulated to reflect the opinion of the authors.
In the abstract our opinion has been highlighted, and so in the section “Making a choice in first line”. The second paragraph, if intended as “First line therapies”, is only descriptive.
2) Table 2, Dimopoulos study on MYD88WT should be reference n°45 and the date 2020 ?
Both references have been added, Dimopoulos 2020 and the abstract with the last follow up of the study (both cohort 1 and cohort 2)
3) Figure 1: should be "CONS" and not "CON" on the left
Corrected
Reviewer 3 Report
The manuscript entitled: “Should patients with Waldenström Macroglobulinemia receive a BTK inhibitor as frontline therapy? by Deodato et al. reviews the impact of BTKis as s frontline therapy in Waldenström patients.
Albeit the paper is well written, prepared and of special interest, some comments should be addressed.
Comments:
1. The title of the review suggest a detailed discussion of BTK inhibitors in Waldenström patients. In the section: “making a choice in first line” the authors should discuss this statement more profoundly.
2. The authors should add more references where appropriate: e.g. line 42, 52, 57, 92,105.
3. Line 60: please defined once in more detail the term: “concomitant medication” and their potential impact on the treatment decision with BTKis.
4. Line 44 space could be deleted.
5. Line 119: why is R-CHOP underlined. Table 1 please delete the yellow highlighted areas where appropriate.
Author Response
Thank you for your comments on our manuscript.
Below is our response to each point.
1) The title of the review suggests a detailed discussion of BTK inhibitors in Waldenström patients. In the section: “making a choice in first line” the authors should discuss this statement more profoundly.
In the section “making a choice in first line” we added a more detailed discussion to answer the question in the title “Should Patients with Waldenström Macroglobulinemia Receive a BTK Inhibitor as Frontline Therapy?”.
2) The authors should add more references where appropriate: e.g. line 42, 52, 57, 92,105.
Appropriate references have been added.
3) Line 60: please defined once in more detail the term: “concomitant medication” and their potential impact on the treatment decision with BTKis.
The term “Concomitant medication” has been defined, and their role on treatment decision has been discussed in the “making a choice in first line” section.
4) Line 44 space could be deleted.
Corrected.
5) Line 119: why is R-CHOP underlined. Table 1 please delete the yellow highlighted areas where appropriate.
Corrected.
Reviewer 4 Report
Deodato et al. wrote the review of treatment option for symptomatic Waldenström macroglobulinemia. They also mentioned the pros and cons of cytotoxic agent with rituximab, proteasome inhibitor-based regimen and BTK inhibitor. The manuscript is well written, informative, and data-driven. Figure regarding pros and cons is simple and visually appealing.
Author Response
Thank you for your kind comment.